# Potential of Cellulose Microfibers for PHA and PLA Biopolymers Reinforcement

**DOI:** 10.3390/molecules25204653

**Published:** 2020-10-13

**Authors:** Gonzalo Mármol, Christian Gauss, Raul Fangueiro

**Affiliations:** 1Centre for Textile Science and Technology (2C2T), University of Minho, 4800-058 Guimarães, Portugal; rfangueiro@dem.uminho.pt; 2School of Science and Engineering, University of Waikato, Hamilton 3216, New Zealand; cgauss@waikato.ac.nz; 3Department of Mechanical Engineering, University of Minho, 4800-058 Guimarães, Portugal

**Keywords:** cellulose, biopolymers, multis-scale, microfibers, cellulose nanocrystals

## Abstract

Cellulose nanocrystals (CNC) have attracted the attention of many engineering fields and offered excellent mechanical and physical properties as polymer reinforcement. However, their application in composite products with high material demand is complex due to the current production costs. This work explores the use of cellulose microfibers (MF) obtained by a straightforward water dispersion of kraft paper to reinforce polyhydroxyalkanoate (PHA) and polylactic acid (PLA) films. To assess the influence of this type of filler material on the properties of biopolymers, films were cast and reinforced at different scales, with both CNC and MF separately, to compare their effectiveness. Regarding mechanical properties, CNC has a better reinforcing effect on the tensile strength of PLA samples, though up to 20 wt.% of MF may also lead to stronger PLA films. Moreover, PHA films reinforced with MF are 23% stronger than neat PHA samples. This gain in strength is accompanied by an increment of the stiffness of the material. Additionally, the addition of MF leads to an increase in the crystallinity of PHA that can be controlled by heat treatment followed by quenching. This change in the crystallinity of PHA affects the hygroscopicity of PHA samples, allowing the modification of the water barrier properties according to the required features. The addition of MF to both types of polymers also increases the surface roughness of the films, which may contribute to obtaining better interlaminar bonding in multi-layer composite applications. Due to the partial lignin content in MF from kraft paper, samples reinforced with MF present a UV blocking effect. Therefore, MF from kraft paper may be explored as a way to introduce high fiber concentrations (up to 20 wt.%) from other sources of recycled paper into biocomposite manufacturing with economic and technical benefits.

## 1. Introduction

Composites are a dominant tool from a material science point of view as their properties facilitate the shape and form design while offering economic advantages for a wide variety of applications. Nevertheless, most of the synthetic petrol-derived composites introduce environmental disadvantages, of which their lack of biodegradability stands out. Plastic-based materials become worthless once they reach the end of their lifespan and, most of the time, these products end up in landfills with a subsequent environmental effect. Using a biobased or biodegradable polymer matrix to manufacture composite materials allows the obtainment of biodegradable products: green composites [1,2]. This technology leads to manufactured goods with a closed lifecycle, which may promote a circular economy. After green composites are discarded from their initial purpose, biogas production may be expected as fuel or feedstock for more biopolymer manufacture [3]. From the different biodegradable bioplastics, polylactic acid (PLA), polyhydroxyalkanoates (PHA), polycaprolactone (PCL), chitosan, starch, and cellulose can be highlighted. However, up to now PLA and PHA are the ones with a wider range of applications given their physicochemical and mechanical properties and reasonable price.

PLA is known as the utmost effective polymer for biomedical and packing composites, and it is the likely choice for the replacement of conventional plastics in a more sustainable way [4]. Nonetheless, other features such as fragility, reduced impact strength, and low thermal stability of PLA become a hurdle for the application of PLA in engineering purposes [5]. Likewise, PHAs are attracting attention as biodegradable alternatives for conventional polymers, and a significant effort has been made in this regard. On the other hand, PHAs are polyesters synthesized by numerous bacteria with remarkable properties for the next generation of environmentally friendly materials [6,7]. Medical and healthcare applications as cardiovascular tissue engineering, cartilage repair, ophthalmological transplantations, nerve regeneration, skin treatment, drug delivery systems, cell anchorage, amongst others are some of the most encouraging uses [6]. It is also possible to find alternative applications of PHA for commercial nonmedical purposes like packaging, fiber material, biofuels, a precursor of carbon material, paper finishing and nanoparticle stabilization amongst others [6]. However, the application of PHA for composite production is challenging due to the associated high cost of its production [8].

Another approach to reducing the composite materials’ carbon footprint is the addition of other biodegradable components in their composition. In this regard, discrete natural fiber reinforcement (NFR) improves the sensitivity to temperature and mechanical performance of PLA, though it is difficult to obtain a homogenous mixture and good adhesion between matrix and fiber. Given the hydroxyl functional groups on the ends of natural fibers that confer them a hydrophilic behavior, there is a polarity variation between the fiber–matrix interface (Figure 1), contributing to their poor adhesion [9]. In the case of PHA matrices, limited mechanical improvement has been reported in the literature when cellulose-based fibers were used as reinforcement [10,11,12].

Since the mechanical properties of composites are essential in material selection, a possible addition with reinforcing properties for both PLA and PHA matrices are cellulose nanoparticles. Several studies reported that the use of cellulose nanocrystals (CNC) in PLA matrices increased the tensile strength [13,14], whereas the addition of CNC in PHA leads to limited reinforcement [15,16]. Besides the improved mechanical properties of the PLA matrices, CNC exhibit singular structural features and excellent physicochemical properties such as biocompatibility, biodegradability, renewability, low density, adaptable surface chemistry, and optical transparency.

However, an aspect that may hinder the addition of CNC in composite commodities is the current production process of this type of filler, which is energy-intensive [17]. Despite all the scientific advances to obtain CNC in a more sustainable way, currently, this type of filler is not suitable for large-scale industrial processes with a high demand for raw materials. 

As an effortless solution to CNC addition, this work explores an innovative processing method to obtain natural fiber reinforcement (NFR) from kraft paper microfibers and their addition in polymeric matrices. The obtainment of this type of fibers is less energy demanding and requires low-tech equipment. Thus, a higher replacement of polymer by natural fibers of up to 30% by mass in composite matrices appears to be possible, bringing both economic and environmental advantages. On one hand, the use of microfibers (MF) from kraft pulp creates a path to incorporate recycled paper in composite manufacture, which increases sustainability and reduces production costs. On the other hand, microfibers (MF) may help to improve the adherence between the polymeric matrix and the natural fiber reinforcement used in later stages for the manufacture of multi-layer green composites, combining several layers of staked biopolymers and natural mats. For that, the production of thin films made out of PLA and PHA will incorporate CNC and MF separately as fillers to assess their comparative tensile strength and microstructural properties.

## 2. Materials and Methods

### 2.1. Materials

Polylactic acid (PLA) Ingeo 4043D was kindly provided by NatureWorks^®^ with an Melt Flow Rate = 6 g/10 min at 10 °C, specific gravity of 1.24 g/cm^3^, and peak melt temperature between 145 °C and 160 °C. Polyhydroxyalkanoate (PHA) was supplied by Goodfellow Cambridge, MFR = 3 g/10 min at 170 °C, specific gravity of 1.24 g/cm^3^ and peak melt temperature between 140 °C and 160 °C. CNC with an average particle size of 75 nm was purchased from Celluforce (Montreal, QC, Canada), in spray-dried form. The paper kraft used in this study is made from commercial Tunisian Alfa bleached *Stipa tenacissasima*. The morphological parameters determined by Morfi apparatus are an average length weighted in length = 841 μm, average width = 16.7 μm, coarseness = 0.073 mg/m, average kink number = 1.253, average angle = 130.8°, kinked fibers = 29.3% and average curl = 8.15%. Reagent grade chloroform (CHCl_3_) with 99% purity was purchase from Honeywell, CAS: 67-66-3, with a density value of 1.48 g/cm^3^ (20 °C) and vapor pressure of 210 hPa (25 °C).

### 2.2. Sample Production

The method to obtain MF from kraft paper started with a shredding process of the kraft until particles of around 20 cm^2^ were obtained. Then paper particles were immersed in water for 48 h. After long water immersion, mechanical stirring was applied for 1 h for paper disintegration. The paper suspension in water was drained and oven-dried at 105 °C until it reached constant weight. To deagglomerate MF clusters, polymer pellets were added to the correspondent amount of MF according to the different formulations (Table 1) and placed in a blender machine for mechanical abrasion between MF and polymer particles.

#### 2.2.1. CNC-Filled Samples

For CNC samples, first 10 mL of CNC suspensions in CHCl_3_ were placed in centrifuge tubes for CNC deagglomeration by mechanical vibration for 10 min using a vortex shaker IKA VORTEX 3. Then, suspensions were diluted by adding CHCl_3_ to obtain a total volume of 50 mL and suspension stabilization was carried out by ultrasonic treatment for 5 min, so proper dispersion and distribution of the fibers in CHCl_3_ was assured. 

#### 2.2.2. MF-Filled Samples

For MF samples, the right fiber dispersion was obtained by high shear mixing using a co-rotating twin-screw extruder Microlab, Rondol^®^. PLA and MF were blended using a temperature profile of 145 °C–200 °C–200 °C–195 °C–190 °C (feed to die) and a rotation speed of 55 rpm, while PHA and MF were blended with a temperature profile of 130 °C–145 °C–150 °C–150 °C–140 °C at a rotation speed of 65 rpm. Later MF-polymer blends were chopped until pellet homogenization. 

For every film production (both CNC-filled and MF-filled samples), 2.84 g of polymer was added to 50 mL of CHCl_3_ (suspension in the case of CNC samples) for its dissolution with the aid of magnetic stirring for 1 h at 60 °C and 1 h cooling down at room temperature. Film casting took place in ceramic trays to promote solvent evaporation at ventilated room conditions. In all the cases, solvent-cast films with a thickness between 0.15 and 0.2 mm were obtained.

### 2.3. Sample Testing

Tensile testing was conducted following the ASTM D882-02 standard. A total of 5 specimens for each formulation were cut into 100 × 10 mm strips and tested at a crosshead displacement of 5 mm/min in a universal testing machine (Hounsfield Tinius Olsen, model H100 KPS) equipped with a load cell of 2.5 kN. The averages of tensile strength, elongation at break and Young’s modulus for each composite formulation are presented with the corresponding coefficient of variation (COV). The differences among the treatment conditions on the evaluated properties were checked by a Tukey test and analysis of variance (ANOVA) for significant (*p* < 0.05) differences. All analyses were performed using MINITAB Release 18 Statistical Software.

Differential scanning calorimetry (DSC) analyses were carried out in a covered aluminum crucible under a nitrogen flow of 100 mL/min in a Metler Toledo, (822^e^) differential scanning calorimeter following the ASTM D3418 standard. Samples were heated from room temperature to 200 °C at a heating rate of 10 °C/min and cooled down at a rate of 10 °C/min down to 30 °C to determine the transition temperature (T_g_), melting temperature (T_m_), cold crystallization temperature (*T_cc_*) and, therefore, calculate the heat of fusion and degree of crystallinity of the samples (*X_c_*). Degree of crystallinity (*X_c_*) was calculated according to [18]:(1)$$X_{c}\left(\%\right)=\frac{∆H_{m}}{W\times ∆H_{polymer}}\times 100$$
where Δ*H_m_* is the melting enthalpy of the samples. Δ*H_polymer_* is the enthalpy for 100% crystalline PLA and PHA, which is approximately 93.6 J/g and 146 J/g, respectively, [18,19] and *W* is the net weight fraction of PLA or PHA in the composites. The enthalpy values were evaluated as the integral of their corresponding peak, i.e., the area under the endothermic peak to estimate the melting enthalpy.

Thermogravimetric analysis was performed using a thermal analyzer (HITACHI STA 7200). Dried samples were introduced in alumina crucibles and were heated from 20 °C to 400 °C at 10 °C/min under nitrogen flow at 40 mL/min.

Infrared spectra of the films were performed on a Shimadzu, IRAffinity-1S Bruker Fourier Transform Infrared Spectrophotometer using an attenuated total reflectance (ATR) module. Transmittance spectra were registered between 400 and 4000 cm^−1^, with a resolution of 4 cm^−1^ and 45 scans.

Microscopic analysis was conducted by optical microscopy, scanning electron microscopy (SEM) and atomic force microscopy (AFM). The surface of the films was assessed by an optical microscope using a Leica DM750M at 50× and 100× magnification. Microfiber length and width, as well as the apparent pore size of the films, were determined for every film formulation by averaging 100 image measurements with the aid of ImageJ software. Morphological analyses were realized in an Ultra-high-resolution Field Emission Gun Scanning Electron Microscopy (FEG-SEM), NOVA 200 Nano SEM, FEI Company. Topographic images were obtained with a secondary electron detector at an acceleration voltage of 10 kV. Before morphological analyses, samples were coated with a thin film (8 nm) of Au-Pd (80–20 weight %), in a high-resolution sputter coater, 208HR Cressington Company, coupled to an MTM-20 Cressington high-resolution Thickness Controller.

The morphological analyses were performed by Scanning Atomic Force Microscopy, using a Nanoscope III Multimode Atomic Force Microscope, from Digital Instruments, where images of 5 × 5 µm^2^ and 10 × 10 µm^2^ were acquired on the surface of all samples. The scanning mode used was intermittent contact or tapping mode, in air, with a cantilever whose spring constant is 42 N/m and the resonance frequency was approximately 310 kHz, determining the surface roughness of the films.

The contact angle was determined after 15 s of the contact of the drop of water with the surface of the film using a Dataphysics OCA Contact Angle System.

The light transmittance of the film samples was determined using a UV–visible spectrophotometer (UV/Vis spectrophotometer), UV-2600 Shimadzu in the wavelength of 200–600 nm. The transmittance at 600 nm (T_600_) and 300 nm (T_300_) was used to evaluate the transparency and UV barrier property of the films, respectively.

## 3. Results and Discussion

The results obtained by the mechanical, physical, and thermal analyses are summarized in Table 2 and Table 3 and discussed in the next sections. 

### 3.1. Tensile Tests

Samples were tested under tensile configuration to obtain the mechanical properties of the different composite formulations. Figure 2 displays the values of tensile strength and elongation at ultimate strength. Comparing the tensile strength values of both neat PLA and PHA samples, PLA exhibits higher values (52.3 MPa) compared to PHA (20.2 MPa). These values are in accordance with the results reported in the literature [20,21]. When CNCs are added to both types of matrices, tensile strength increases for every formulation. The greater improvement in tensile strength is most noticeable in PLA samples, where a maximum increase of up to 38% was achieved for 1% CNC PLA samples, with no evident effect on the elongation at break. In fact, all the PLA formulations reinforced with CNC presented statistically equivalent elongation at break values (Table 2). Regarding PHA samples, for every CNC addition content, a lower tensile strength enhancement (approximately 10%) is observed compared to PLA films. However, the elongation at break is considerably improved, especially the 1% CNC PHA formulation with an increase of approximately 130%.

Regarding the use of MF as filler, PLA samples have lower tensile strength compared to PLA containing CNC. Samples with 30% by mass of MF show a decrease in tensile strength. However, tensile strength improves by around 25% with preservation of the elongation at ultimate strength in the 20% MF-PLA samples. Therefore, the specific energy absorbed by 20% MF-PLA samples increases under tensile loading. In comparison with samples with CNC, tensile strength improves even more for MF-PHA composites, with an increase between 14 and 23%, where the highest value is obtained by the 20% MF samples. Nevertheless, this increase in tensile strength is accompanied by a reduction of the elongation at ultimate strength.

Some studies indicate that the addition of fibers to non-polar thermoplastics reduces the tensile strength due to the poor fiber–matrix adhesion at the interface, which is related to low interfacial stress transfer [22]. In order to improve the reinforcing effect of the fibers in these polymers, coupling agents are used to increase the compatibility between the fibers and the matrix [3]. However, in this work, MF addition (without a coupling agent) led to enhancements in both tensile modulus and strength, showing affinity between the MF and the polymers used here (PLA and PHA). Consistent with these findings, previous studies using PHA composites reinforced with agave fibers presented identical results, where the tensile modulus of the PHAs increased with fiber addition without the use of coupling agents [23]. Similarly, PLA-based composites reinforced with regenerated cellulose (Lyocell), hemp, kenaf, or cotton also present an improvement in tensile strength and Young’s modulus without the addition of coupling agents [24]. 

For all the reinforced samples produced in this study, Young’s modulus increases compared to neat polymer films (Table 2), though in some cases without a statistical difference, with the exception of 3% CNC-PLA samples that exhibit slightly lower stiffness values compared to neat PLA samples. The same positive behavior is observed on the tensile strength of all the reinforced composites, except the samples of PLA reinforced with 30% of MF. Thus, it is proven that the addition of MF in both types of biopolymer may present a reinforcing effect when adequately formulated.

In tensile tests that used non-woven flax mat PHB composites produced by the film-stacking method with different fiber contents, the stiffness of the composite materials increased with the fiber content [25]. For high fiber contents (>30 vol.%), flax/PHA composites presented similar elastic modulus to short fiber flax/polypropylene and glass-mat-reinforced thermoplastic composites. Regarding tensile strength, the effect of fiber addition to PHA was not significant, whereas the addition of flax resulted in a lower elongation at ultimate strength of the composites, i.e., approx. 1.5% for all fiber volume fractions. Thus, the results brought here suggest a favorable application of MF to reinforced PHA matrices in composite products.

### 3.2. Differential Scanning Calorimetry

Differential scanning calorimetry tests were performed to evaluate and compare the effect of the reinforcement addition on the crystallinity of each polymer. Since annealing, i.e., slow cooling after molding, relieves the internal stresses introduced during the film fabrication [26], a thermal treatment consisting of a heating at 80 °C for 24 h and later cooling at room temperature was applied before DSC testing. The temperature chosen lies in between the glass transition temperature, T_g_ (~60 °C) and the cold crystallization temperature, T_cc_ (>100 °C). It has been found that the polymer crystallite morphology can be also modified upon annealing at a temperature higher than the T_g_, thus changing the physical properties of the polymers [26]. Thermal annealing is a simple route for stabilizing glassy polymers via the densification of their polymer chains. The effects of annealing are depicted on the first DSC heating cycle of the samples (Figure 3a). Since no exothermic peak was observed during cooling, the ΔH_m_ value was used to calculate the X_c_ of the PLA/PHA–CNC/MF composites (Table 3). Compared to the neat PHA pellets, pure PHA films after solvent casting undergo a reduction in crystallinity, while the reinforcement of the films leads to higher crystallinity regardless of the type of reinforcement. For low X_c_ values (Table 3) as is the case of PHA samples (8.1%), the crystalline nature of the reinforcements may lead to an increase in the crystallinity nucleation of the matrix due to a better arrangement of the polymer around the surface of the fillers [27]. Therefore, higher reinforcement contents have a more pronounced effect on polymer crystallinity, since MF addition has a higher impact on the crystallinity of PHA compared with CNC samples. On the other hand, PLA films increase their crystallinity with annealing in comparison with samples without thermal treatment. The addition of the reinforcement reduces the crystallinity of the films for most formulations, although the highest crystallinity is achieved by samples with 10% MF. A single peak in the melting region of neat PLA (Figure 3a) indicates a homogeneous distribution of ordered crystals. However, the addition of MF in PLA matrices leads to a heterogeneous crystal distribution, evidenced by a two-step melting process.

Since a second heating step at a rate of 10 °C/min was applied after a fast cooling (quenching), the effect of quenching on the crystallinity of the samples is expected to be observed (Figure 3b). The quenched PLA samples exhibit a strong cold crystallization peak, which is particularly sharp for MF reinforced PLA films. In contrast, PHA samples exhibit a two-step broader low intensity cold crystallization peak compared to the first heating. Thus, quenching leads to a recovery of the crystallinity of PHA samples, to values similar to the initial values before processing (Table 3), and increases T_cc_, though a heterogeneous crystal distribution is promoted [28]. These exothermic peaks reveal the presence of a large number of active nuclei. For neat PLA films, quenching reduces crystallinity, but for PLA samples reinforced with MF, particularly for higher concentrations (20 and 30%), this fast cooling is translated into a higher crystallinity.

### 3.3. Fourier Transformed Infrared Spectroscopy

In order to assess any possible chemical interaction between the different polymers and the reinforcing filler, FTIR analysis was performed. Regarding PHA samples, the bands in the range 3015–2955 cm^−1^ are assigned to –CH_3_ asymmetric stretching vibrations and those in the range 2940–2915 cm^−1^ to –CH_2_ asymmetric stretching vibrations [29]. The range from 2885 to 2845 cm^−1^ is distinctive of symmetric stretching modes of –CH_3_ and –CH_2_ [29,30]. The peaks at 2976 cm^−1^, 2934 cm^−1^ and 2874 cm^−1^ and the shoulder at 2923 cm^−1^ arise from the crystalline state and that at 2997 cm^−1^ from the amorphous phase [31,32]. The shoulder at 3007 cm^−1^, associated to –CH_3_ asymmetric stretching, points to the presence of intermolecular CH–O hydrogen bonds in PHB crystals [29]. Regarding the composites’ spectra, little differences may be noticed when composites are reinforced with CNC due to the low concentration of the fillers and, therefore, no chemical modification is perceived in the functional groups. In the case of MF reinforced composites, at 1163 cm^−1^ there is an increase in intensity that represents the –C–O–C– stretching of the cellulose of the reinforcement [33]. However, additional functional groups are not observed, evidencing a lack of chemical interference. 

From the FTIR spectrum of PLA samples (Figure 4b), distinct characteristic FTIR peaks were observed at 866 cm^−1^, 1073 cm^−1^, 1454 cm^−1^, 1742 cm^−1^ and 2926 cm^−1^ corresponding to –C–O–C– bond stretching, –CH_3_ asymmetric vibrations, –CH bending vibrations, –C=O vibrations and –CH_3_ symmetric vibrations, respectively, for neat PLA films [34]. All these peaks are also present in every PLA-based sample, which indicates that PLA functional groups are not altered with the addition of reinforcing fillers. For the case of CNCs, the peak at 2808 cm^−1^ represents the –CH stretching band, peaks at 1163 cm^−1^ represent the –C–O–C– stretching in β-1,4-d glycosidic linkage present in CNCs and peaks at 1427 cm^−1^ represent the symmetric –CH_2_ bending peak [33]. The presence of peaks near 1750 cm^−1^ in all the composites indicates the presence of free –CO groups. Similarly, the peaks around 1450 cm^−1^ and 2940 cm^−1^ in all the nanocomposites represent the symmetric –CH_2_ bending and stretching vibrations, respectively [33]. It is worth mentioning that the spectrum of MF is similar to CNC, revealing a high cellulose content in the microfibers after their bleaching.

### 3.4. Thermogravimetric Analysis

Figure 5 shows the thermal degradation with the increase in temperature of each analyzed polymer, reinforcement filler, and composite to assess and compare the influence of the addition of cellulose particles in these polymeric films. Regarding the different types of reinforcement, MF presents a higher peak degradation temperature (357 °C) compared to CNC (305 °C). This difference may be attributed to a partial lignin content in MF even after their bleaching since lignin undergoes thermal degradation up to 600 °C [35]. PHA (Figure 5a) exhibits a two-step degradation process, with two clearly defined peaks centered at 275 °C and 345 °C. The thermal decomposition between 170 °C and 250 °C is related to the loss of low molecular weight compounds of the biopolymer [36]. The maximum mass loss of PHA occurs at 275 °C and is associated with the ester cleavage of the PHA component by elimination reaction [36]. The addition of both MF and CNC reinforcement increases the thermal stability of PHA since the mass loss is reduced in the composites at 275 °C. The difference in behavior between MF and CNC reinforced composites at 345 °C is related to the mass fraction of MF-based samples, which is 10 times higher than the filler content in CNC-based samples. However, the overall thermal stability is similar for samples reinforced with low MF content (10 wt.%) compared to CNC samples. In the case of PLA films, pure PLA samples present a single-step decomposition process with a maximum decomposition rate at 375 °C, which takes place at higher temperatures than in the PHA samples (275 °C) and both pure CNC (305 °C) and MF (357 °C) (Table 3). The PLA degradation reaction is based on a hydroxyl end-initiated ester interchange process and chain homolysis [37]. Contrary to what is noted for PHA samples, the addition of MF in PLA samples (Figure 5b) reduces the thermal stability of the films by 37%. However, the addition of CNC preserves the thermal stability of the PLA films, since it has little effect in this regard given the low CNC content.

### 3.5. Microscopy Analysis

From Figure 6a,d it can be noted that both PHA and PLA films present a smooth surface with small apparent pores, 22 and 27 μm, respectively. This apparent porosity is caused by the solvent casting method since during solvent evaporation some air bubbles are entrapped and remain as voids within the solid phase. Despite the porosity of the samples, the neat polymer matrices exhibit conventional mechanical performance compared to the results reported in the literature [20,21]. A clear effect of the addition of CNC in both PHA and PLA is the increase in apparent porosity since for PHA samples the number of superficial pores is increased (Figure 6b) and for PLA samples the size of the pores is enlarged (Figure 6e). This rise in apparent porosity leads to higher specific surfaces, which translate into modifications of the hygroscopy of the materials.

Contact angle measurement was conducted to evaluate the wettability of the pristine PHA and PLA film and the nanocomposite films after their combination with different concentrations of CNC and MF. Contact angle measurements and surface roughness may be related to each other [27]. Contact angle results (Table 3) reveal that the hydrophobicity, and therefore the water barrier properties of the nanocomposite film, are somewhat related to the crystallinity degree of the films, rather than the concentration of fillers. When the X_c_ is compared with the contact angle for every sample after the first heating stage, a higher crystallinity degree leads to higher hydrophobicity for both types of polymer. The increase in crystallinity of PHA films with the addition of both types of reinforcement is also reflected in the increase in the contact angle. It might be explained by the crystalline nature of the reinforcements, which leads to an increase in the hydrophobicity of the matrix, thus increasing the contact angle [27]. Therefore, the water barrier properties of this type of films may be functionalized by thermal treatment according to the desired properties. Thus, when MF is added to both types of biopolymers, PLA and PHA, a more hydrophobic performance may be achieved by film quenching.

In the case of samples reinforced with MF, a rough surface with a dense network of microfibers surrounded by a thin coating of polymer is observed (Figure 6c,f). The average length of the fibers after casting is 332.2 μm with an average width value of 11.3 μm. The length of the fibers decreases during the high-shear mixing process compared to the initial morphological characterization. The increase in apparent porosity is also confirmed by SEM analysis. Figure 7 shows the surface of PHA films with no reinforcement (a), with 1% CNC (b) and with 20% of MF. In spite of reducing the number of apparent pores, it is clear that the addition of CNC increases the pore size compared to unreinforced PHA, from an average pore size of 2.2 μm up to a 7.1 μm. A similar trend is observed in PLA samples, where apparent pore size increases from 2.7 μm up to 8.1 μm. With the addition of MF, the number of pores remains at the same level as for unreinforced samples, while the average pore size increases up to 3.8 μm.

The increase in apparent pore size for PHA is accompanied by an increase in roughness, as is shown by AFM in Figure 8. The different mean roughness values for unreinforced PHA, 1% CNC and 20% MF reinforced samples are 117.7 nm, 189.6 nm and 318.7 nm, respectively. Surface roughness is an essential parameter for multi-layer composites since it may affect the interfacial bonding between the polymer and reinforcement layers. Surface roughness promotes an enhancement of the interfacial adhesion and allows greater stress transfer between the matrix and reinforcements, reducing the capacity of fiber debonding. Therefore, the increase in surface roughness with the addition of MF may benefit the application of this type of material in multi-layer composites.

AFM also confirms the modification of the surface roughness with the addition of fibers for the PLA composite samples (Figure 8). As noted for PHA, in PLA films the roughness is also increased, from 74.4 nm (plain PLA) up to 88.7 and 174.8 nm for 1% CNC and 20% MF samples, respectively. Once again, the gain in roughness is more notorious for samples reinforced with MF, which can be translated into a better interlaminar bonding in multi-layer composite applications. In the case of PLA samples, SEM (Figure 7) reveals that fractured surfaces, even after their necking, are rougher with the addition of different reinforcements. In Figure 7d, a neat PLA sample of around 2 μm thickness (cross-section area is around 100 times smaller than the average produced films with 200 μm thickness) presents a smooth surface, while Figure 6e,f clearly shows rougher surfaces.

### 3.6. Ultraviolet-Visible Spectroscopy

In the case of UV protective films (UV-PF), the light-resistant capacity of the UV-PF is the main indicator of packaging materials for UV susceptible products. In this study, fiber-reinforced thin transparent films produced by solvent casting were investigated by light transmittance analysis (Figure 9). PLA films showed no UV light transmission in the lower range of UV-C (100–230 nm) and started to show transmission (up to 40%) in the higher range of UV-C (230–280 nm) (Figure 9a). From this point, the transmittance of PLA increases to ~90% for UV-B (280–315 nm) and 315–400 nm (UV-A), which remained constant in the wavelength range of 350–800 nm. The addition of CNC to PLA films has an insignificant effect on the PLA film transmittance in this region. For wavenumbers below 350 nm, the addition of CNC reduces the transmittance by around 5% for the higher concentration contents (2 and 3 wt.%). In contrast, all PLA-MF composites showed decreased transmittance across all wavelengths. This variation in transmittance is related to the presence of UV-absorbing chromophores in the reinforcing particles of MF. The lignin content of the MF may explain this effect, as observed in the thermogravimetric analysis [35]. Lignin is well known to be UV absorbent because of the presence of UV-absorbing chromophores such as phenolic and ketone groups [38,39,40]. Therefore, MF-reinforced PLA films were able to absorb UV, and the higher the contents of MF in the film (10, 20 and 30 wt.%), the higher the UV absorption of the composite, displaying transmittance values of 79.1, 70.3 and 40.3%, respectively. 

Regarding neat PHA samples presented in Figure 9b, the transmittance values are lower compared to PLA films, with values of around 70% at 600 nm. In contrast to PLA films, the addition of high concentrations of CNC (2 and 3 wt.%) in PHA specimens leads to a 15% decrease in the transmittance through the entire spectrum. The major difference between PHA and PLA films is the improvement in the UV light barrier ability of the PHA films. This UV blocking effect is more pronounced when MF is added, in comparison to samples reinforced with CNC. At 600 nm, when MF is added in different concentrations (10, 20 and 30 wt.%) in PHA, the transmittance is reduced by 1.1, 33.1 and 244%, respectively. This UV blocking effect is more evident for shorter wavelengths. These results indicate that both PHA and PLA composites reinforced with MF present better UV-protecting properties compared to samples with CNC.

## 4. Conclusions

The addition of MF cellulose-based fibers is presented as an alternative to reinforcing biopolymers films, offering some advantages compared to the addition of CNC. Despite the higher specific surface area of CNC and, therefore, the advantages associated with it, it also reduces the possibility of introducing higher concentrations of this type of filler in composite applications, especially in commodity ones. Moreover, the high production cost of CNC also limits the application of this type of filler for high added value purposes, whereas MF from kraft pulp may help to reduce not only the cost of the filler but the whole composite with the addition of up to 20 wt.% of reinforcement.

Compared to CNC, MF have a better reinforcing effect on PHA samples, increasing both tensile strength and Young’s modulus. Combined with this improvement of the mechanical performance of PHA films, the addition of MF causes a notorious increase in the roughness of the films, compared to both neat PLA and PHA, and composites reinforced with CNC. This added roughness may have interesting applications in multi-layer composites where improved interlaminar bonding is required. Moreover, the surface affinity of the films to water may be controlled by the addition of MF and specific cooling conditions since the crystallinity degree of the film is related to the contact angle with water. In addition, the use of MF as reinforcement in biopolymer films may be applied in UV-blocking elements since MF clearly diminishes UV-VIS transmittance in the whole wavelength range.

After this overall characterization of biopolymer films reinforced with MF, a more in-depth characterization should be addressed in order to better understand the performance of this type of reinforcement for specific applications. Considering the benefits of both types of reinforcements explored here, CNC and MF, a possible hierarchical reinforcement can be conducted to understand their synergic effect on the properties of this type of composite. Moreover, possible treatments on cellulose fibers would introduce additional functionalities and improved properties to bio-based composites.

Along with the improved mechanical properties of the biopolymers evaluated in this work with the addition of MF, the use of MF brings an important economic advantage. This reduction in cost is related to the price of this type of filler. In this work, the authors used commercial kraft paper to avoid any heterogeneity in the quality of the microfibers and, therefore, to obtain consistent films. Even using this high-quality raw material, the price of the filler is much cheaper compared with PHA. Then, this filler, when dosed at 30 wt.%, clearly reduces the cost of the polymer composite. However, the great advantage of using MF is the possibility of reusing recycled kraft paper once is discarded from its initial purpose. Recycled kraft paper from a selective waste collection after its processing could be up to 10 times cheaper than technical grade kraft paper.

It is remarkable that the defibrillation process applied in this study only involves immersion in water, low-energy mechanical stirring and drying. In the end, in industrial processes, up to 90% of the energy demand of this process is due to water evaporation. Using efficient drying procedures such as Through Air Drying (TAD), water removal involves 4800 kJ/kg of water evaporated [41]. This requires around 695 kWh to obtain a ton of dried microfibers. In addition, the water required in this process could be used in a loop cycle after decontamination (impurities, oil and pigments) through straightforward flocculation/filtration techniques.

## Figures and Tables

**Figure 1 molecules-25-04653-f001:**
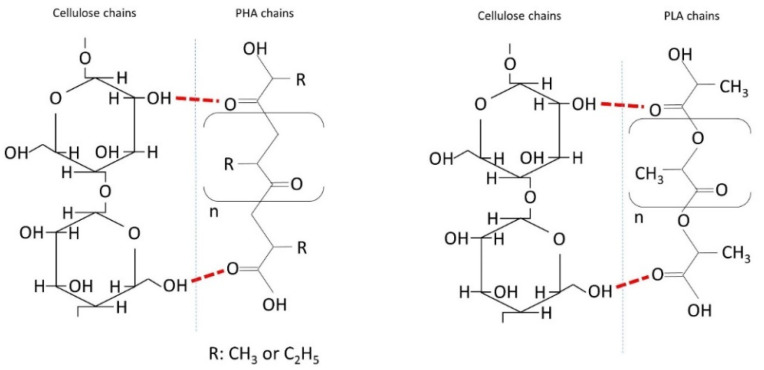
Schematic connection of polylactic acid (PLA) and polyhydroxyalkanoate (PHA) with cellulose.

**Figure 2 molecules-25-04653-f002:**
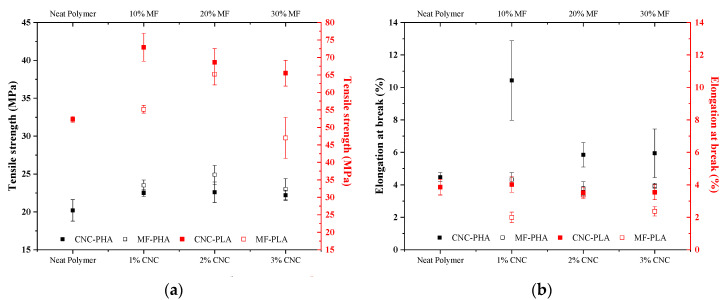
Tensile strength (**a**) and elongation at break (**b**) of PHA and PLA films reinforced with CNC and MF with varying the filler ratios. Error bars indicate one standard deviation in all cases.

**Figure 3 molecules-25-04653-f003:**
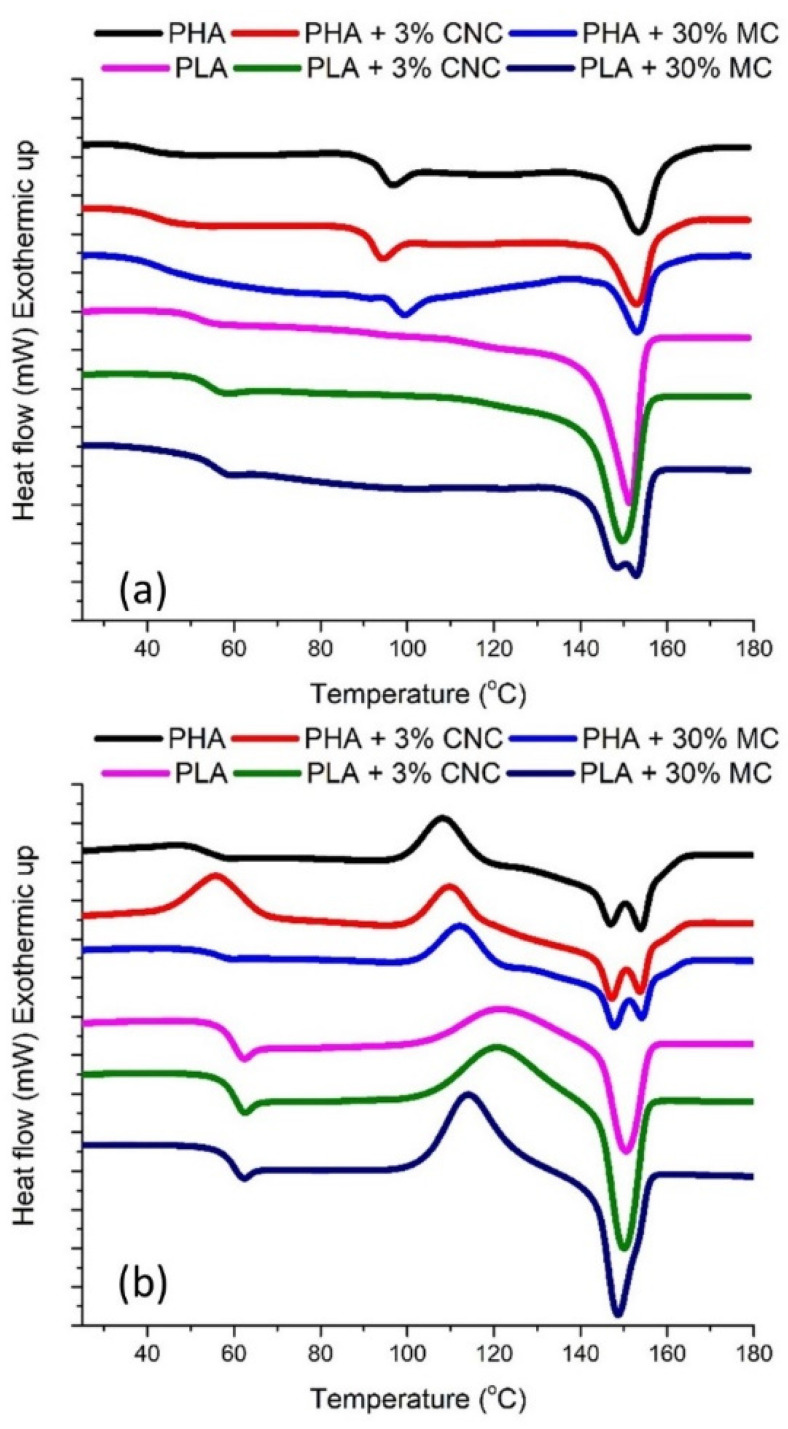
Differential Scanning Calorimetry curves of the PHA and PLA films reinforced with CNC and MF during the first heating stage (**a**) and during the second heating stage (**b**).

**Figure 4 molecules-25-04653-f004:**
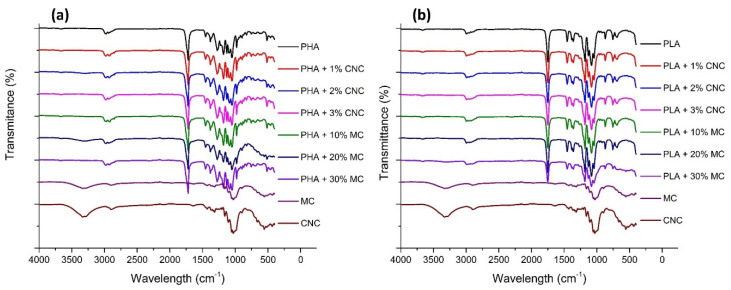
Fourier Transform Infrared spectrograms of the PHA (**a**) and PLA (**b**) films reinforced with CNC and MF.

**Figure 5 molecules-25-04653-f005:**
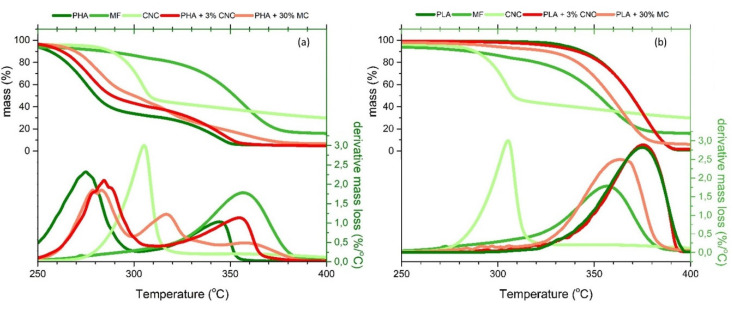
Thermograms and their derivatives of the PHA (**a**) and PLA (**b**) films reinforced with CNC and MF.

**Figure 6 molecules-25-04653-f006:**
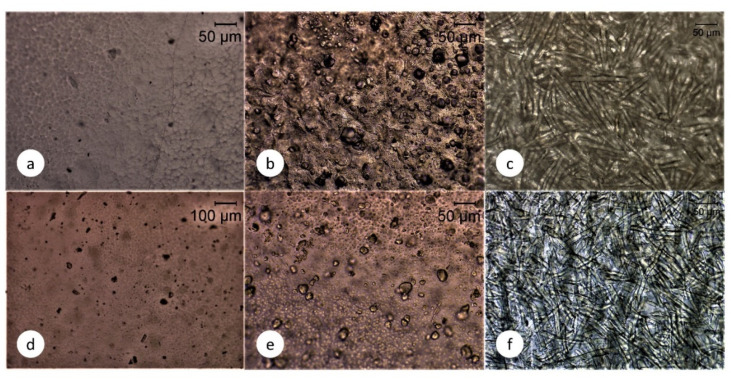
Surface optical micrographs of neat PHA films (**a**), PHA films reinforced with 2 wt.% CNC (**b**), reinforced with 20 wt.% MF (**c**), neat PLA (**d**), PLA films reinforced with 2 wt.% CNC (**e**), reinforced with 20 wt.% MF (**f**).

**Figure 7 molecules-25-04653-f007:**
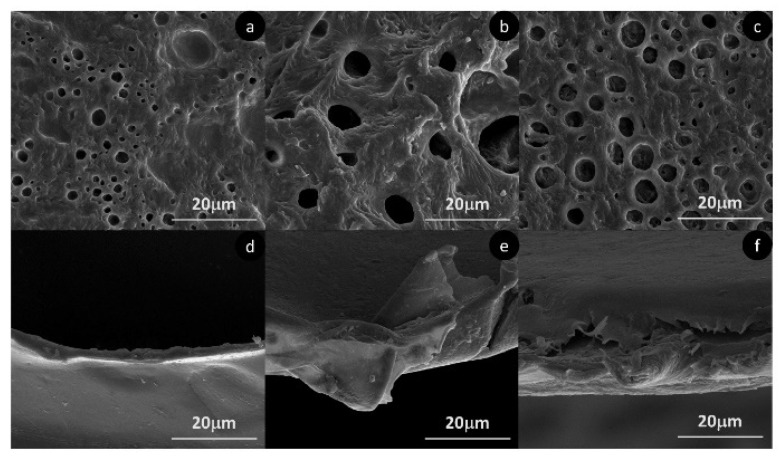
Scanning electron microscope of films surface of neat PHA (**a**), PHA + 2% CNC (**b**), PHA + 20% MF (**c**) and fracture surface of neat PLA (**d**), PLA + 2% CNC (**e**) and PLA + 20% MF (**f**).

**Figure 8 molecules-25-04653-f008:**
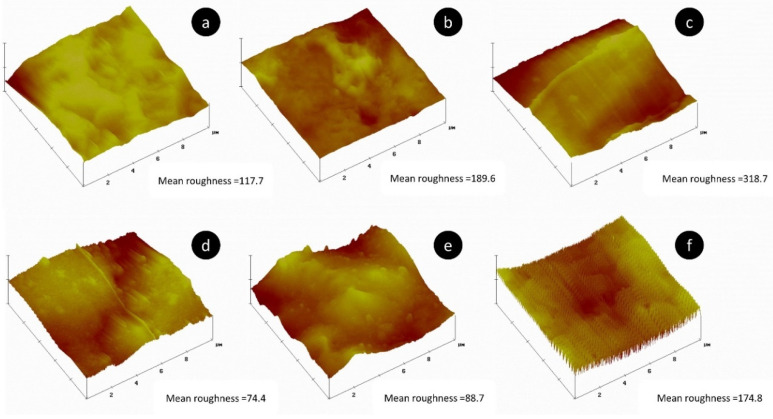
Atomic Force Microscopy height map diagram and mean roughness values of neat PHA (**a**), PHA + 2% CNC (**b**), PHA + 20% MF (**c**), neat PLA (**d**), PLA + 2% CNC (**e**) and PLA + 20% MF (**f**).

**Figure 9 molecules-25-04653-f009:**
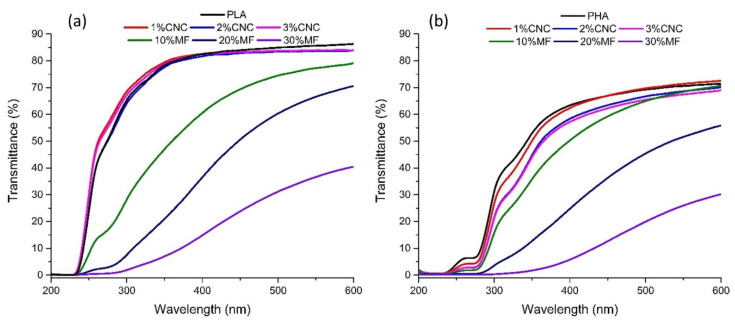
UV-Vis spectrograms of the PHA (**a**) and PLA (**b**) films reinforced with CNC and MF.

**Table 1 molecules-25-04653-t001:** Formulations of the different films assessed.

	Polymer	Filler Type	Content (wt.%)
PHA	PHA	None	0
PHA + 1% CNC	PHA	CNC	1
PHA + 2% CNC	PHA	CNC	2
PHA + 3% CNC	PHA	CNC	3
PHA + 10% MF	PHA	MF	10
PHA + 20% MF	PHA	MF	20
PHA + 30% MF	PHA	MF	30
PLA	PLA	None	0
PLA + 1% CNC	PLA	CNC	1
PLA + 2% CNC	PLA	CNC	2
PLA + 3% CNC	PLA	CNC	3
PLA + 10% MF	PLA	MF	10
PLA + 20% MF	PLA	MF	20
PLA + 30% MF	PLA	MF	30

Sample preparation for each type of filler used, either cellulose nanocrystals (CNC) or microfibers (MF), followed different routes.

**Table 2 molecules-25-04653-t002:** Mechanical properties PLA and PHA composites with different reinforcing fillers (CNC and MF). Same letters (a, b, or c) mean there is no statistical difference among the composites with the same matrix.

*n* = 5	Tensile Strength (MPa)		Elongation at Break (%)		Young’s Modulus (GPa)	
	Avr.	COV	Avr.	COV	Avr.	COV
PHA	20.2 ^c^	0.07	4.47 ^b^	0.06	0.54 ^c^	0.12
PHA + 1% CNC	22.5 ^a,b,c^	0.02	10.43 ^a^	0.23	0.72 ^b^	0.02
PHA + 2% CNC	22.6 ^a,b,c^	0.06	5.85 ^b^	0.13	0.81 ^a,b^	0.06
PHA + 3% CNC	22.2 ^b,c^	0.03	5.95 ^b^	0.25	0.81 ^a,b^	0.05
PHA + 10% MF	23.5 ^a,b^	0.03	4.33 ^b^	0.10	0.81 ^a,b^	0.03
PHA + 20% MF	24.9 ^a^	0.05	3.78 ^b^	0.11	0.94 ^a^	0.14
PHA + 30% MF	23.0 ^a,b^	0.06	3.93 ^b^	0.04	0.93 ^a^	0.06
PLA	52.3 ^b,c^	0.02	3.86 ^a^	0.13	1.95 ^b,c^	0.05
PLA + 1% CNC	72.9 ^a^	0.05	4.02 ^a^	0.12	2.12 ^a,b,c^	0.07
PLA + 2% CNC	68.6 ^a^	0.06	3.51 ^a^	0.10	2.05 ^a,b,c^	0.03
PLA + 3% CNC	65.5 ^a^	0.06	3.54 ^a^	0.13	1.92 ^c^	0.03
PLA + 10% MF	55.2 ^b^	0.02	2.00 ^b^	0.16	2.28 ^a,b^	0.10
PLA + 20% MF	65.2 ^a^	0.05	3.38 ^a^	0.04	2.39 ^a^	0.09
PLA + 30% MF	47.0 ^c^	0.12	2.37 ^b^	0.12	2.36 ^a^	0.07
MF	Fiber length			841 ± 71 μm		
CNC				≈75 nm		

**Table 3 molecules-25-04653-t003:** Summary of results obtained by physical characterization of PLA and PHA composites with different reinforcing fillers (CNC and MF). Different letter between treatments means a significant statistical difference (*p* > 0.05).

	Maximum Degradation Rate				First Heating			Second Heating				UV-VIS Transmittance	
	Temperature at Peak Center		Mass Loss at Peak Center		X_c_	T_g_	T_cc_	X_c_	T_g_	T_cc_	Contact Angle	T_300_	T_600_
	(°C)		(%)		(%)	(°C)		(%)	(°C)		(°)	(%)	
PHA pellets					12.7	60.4							
PLA pellets					28.5	60.8							
PHA	269	342	39.81	87.86	8.1	**	96.8	11.6	48.6	108.5	55.7	31.26	71.6
PHA + 1% CNC	270	330	35.16	88.38	7.2	**	94.4	12.8	61.8	110.7	73.8	26.82	72.5
PHA + 2% CNC	273	330	32.45	87.93	8.5	**	94.2	12.2	48.6	110.2	95.7	21.72	70.1
PHA + 3% CNC	275	317	29.93	87.15	8.3	**	94.1	11.8	55.4	109.8	78.8	21.22	69.0
PHA + 10% MF	270	341	35.11	91.21	9.1	**	96.8	13.9	49.2	112.3	78.6	15.7	71.0
PHA + 20% MF	277	341	19.08	80.55	10.4	**	95.7	12	51.4	112.3	78.9	3.59	55.6
PHA + 30% MF	278	341	18.14	88.03	7.8	**	99.6	13.5	52.8	113.4	67.5	0.39	30.1
PLA	368		57.46		42.7	55.6	**	26.1	62.7	121.6	67.5	64.66	86.3
PLA + 1% CNC	368		54.19		33.1	57.7	**	25.2	63	123.8	43.6	68.07	84.2
PLA + 2% CNC	368		54.13		34.4	57	**	19.5	62	120.5	57.2	63.76	83.8
PLA + 3% CNC	368		54.06		37.2	57.7	**	37.6	62.7	124.3	60.6	66.71	83.8
PLA + 10% MF	360		74.88		45.6	57	**	50.7	62.5	118.2	67.6	27.81	79.1
PLA + 20% MF	357		77.6		38.9	59.4	**	59.9	62.5	118	61.1	7.01	70.3
PLA + 30% MF	357		77.24		30.7	59.5	**	31.2	62.4	114.2	60.9	2.04	40.5
MF	357		8.14		Fiber length			841 ± 71 μm					
CNC	305		4.67					≈75 nm					

** Stand for non-determined values.

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
