# Peer review of "Potential of Cellulose Microfibers for PHA and PLA Biopolymers Reinforcement"

_molecules, 2020, doi:10.3390/molecules25204653_

Round 1

Reviewer 1 Report

Submitted manuscript entitled “Potential of cellulose microfibers for PHA and PLA
biopolymers reinforcement” describe fabrication of new materials.
The authors present preparation and characterization using wide range of the
complementary methods among others: DSC, TGA, FTIR, SEM, AFM.
The work contains the latest literature review on the topic in question. My
recommendation is that the article needs a minor revision.
Comments:
1.Graphical abstract should be added
2. Schematic connection of PLA and PHA with cellulose should be added.
3. Was the elemental analysis performed for discussed compounds? If so, it would
be beneficial to include these data.
4. In submitted manuscript there are numerous notes: “Error! Reference sour- ce
not found ”, please correct it.
5. An extensive literature review has been presented by the authors however a
specific technological application for synthesized compounds is not mentioned at
all. Is it the purely scientific research of new materials or maybe submitted article is
connected strictly with the search of the materials with well-defined
physicochemical properties.
6. Figure 6.- Too small markings on images (measurement parameters).
7. How was the doping percentage chosen, based on previous literature data?
8. What is the thickness of the films to be potentially used in industrial applications?
9. How the film thickness influences on the physical and chemical properties ?
10. More information regarding implementing the project in an industrial scale
should be given. Is the process ecological and economic from the financial po- int
of view? Are there any benefits comparing to current/existing solutions?

Author Response

Comments:
1.Graphical abstract should be added

A graphical abstract has been added as requested.

2.Schematic connection of PLA and PHA with cellulose should be added.

A schematic connection of PLA and PHA with cellulose has been created. It has been listed as Figure 1.

3.Was the elemental analysis performed for discussed compounds? If so, it would
be beneficial to include these data
.

Elemental analysis was not performed as all the available information used in this study was provided from the different commercial providers.

4.In submitted manuscript there are numerous notes: “Error! Reference source not found ”, please correct it.

This error must have been originated by the manipulation of the original document because of incompatibility of text processor between the authors and edition staff. All these notes are related to Microsoft Word Cross References and have been addressed.

5.An extensive literature review has been presented by the authors however a specific technological application for synthesized compounds is not mentioned at all. Is it the purely scientific research of new materials or maybe submitted article is connected strictly with the search of the materials with well-defined physicochemical properties.

This work was intended to explore different applications for these types of biopolymer where great amount of plastic is required. This is the case of multi-layered structural composites. One of the possible benefits derived from this study is the improvement of the adherence between the polymeric matrix and the natural fiber reinforcement used in later stages for the manufacture of multi-layer green composites, combining several layers of staked biopolymers and natural reinforcing mats. This has been highlighted in the introduction. However, this aspect has not been further discussed here since ongoing work is dealing with this concern.

6.Figure 6.- Too small markings on images (measurement parameters).

Figure 6 has been reedited. Since the measurement parameters were illegible and added minor information, they have been removed and the scale bar has been relocated.

7. How was the doping percentage chosen, based on previous literature data?

The mass fractions of MF (up to 30wt.%) were chosen since most of the literature traditionally work in this range of micro filler addition. Besides, higher MF mass fractions, given the physical properties of this type of filler (high surface area), hindered the feeding of this type of filler, so homogenization of the extruded material was not achieved. Mass fractions below 5wt.% were initially intended. However, the produced formulations at these reduced filler content could not be accurately controlled, so insufficient accuracy was achieved for conclusive results. Moreover, this research was looking for high polymer replacement to reduce the cost of the biopolymer composites.

Similarly, CNC shows the best mechanical performance in the range of 1-2 wt% according to the literature.

8.What is the thickness of the films to be potentially used in industrial applications?

Following the emphasized principle to answer question #5, one of the potential industrial applications for this type of material is the use in multi-layer green composites with short life-spam, in which recyclability can play a key role once the composite is discarded. For this application, where tensile strength and Young’s modulus is increased with the addition of MF in PHA, the composite would be applied as a matrix layer. The total thickness of the layer of the matrix can be obtained, after optimization of the mass fraction between the layers of matrix and natural fabric reinforcement, by stacking several layers as these here produced.

9.How the film thickness influences on the physical and chemical properties?

The target thickness used in this work (0.15-0.20 mm) matches the average thickness of the polymer films conventionally produced in industrial processes. Besides this, the thickness here aimed was the optimal for a homogeneous film structure using the solvent casting method. Thus, the influence of the film thickness on the physical and chemical properties is out of the topic here addressed.

According to the literature, when the film thickness is reduced the Young’s modulus decreases too, while the optical transmittance is enhanced. However, this factor was not considered for this work.

10.More information regarding implementing the project in an industrial scale should be given. Is the process ecological and economic from the financial point of view? Are there any benefits comparing to current/existing solutions?

Along with the improved mechanical properties of the biopolymers evaluated in this work with the addition of MF, the use of MF brings an important economic advantage. This reduction in cost is related to the price of this type of filler. In this work, the authors used commercial kraft paper to avoid any heterogeneity in the quality of the microfibers and, therefore, to obtain consistent films. Even using this high quality raw material, the price of the filler is around $500-1500 per metric ton, which is much cheaper compared with PHA, with a cost of around $2500-4500 per metric ton. Then, this filler, 3-5 times cheaper than the polymer, when dosed at 30 wt.%, reduces the cost of a polymer matrix in between 10 and 15%. However, the great advantage of using MF is the possibility of reusing recycled kraft paper once is discarded from its initial purpose. Recycled kraft paper from selective waste collection after its processing could be up to 10 times cheaper than technical grade kraft paper.

It is remarkable that the defibrillation process applied in this study only involves immersion in water, low-energy mechanical stirring and drying. In the end, in industrial processes, up to 90% of the energy demand of this process is due to water evaporation. Using efficient drying procedures such as Through Air Drying (TAD), water removal involves 4800 kJ/kg of water evaporated. This requires around 695 kWh to obtain a ton of dried microfibers. In addition, the water required in this process could be used in a loop cycle after decontamination (impurities, oil and pigments) through straightforward flocculation/filtration techniques.

This has been included in the final conclusions of the work

Reviewer 2 Report

The novelty of paper is not new and cannot convey any specific information to this field of science.

Author Response

The novelty of the paper is not new and cannot convey any specific information to this field of science

We respect the opinion of the reviewer. However, this paper brings two important features to implement the use of bioplastics in a large scale. On one side, the production method here exposed to obtain cellulose microfibers is low-energy demanding and allows the reinforcement of PHA with high cellulose content. This production method allows reducing the cost of the films as well as improving the mechanical properties. The second facet is the possibility of applying this procedure to recycled paper, which would promote a more efficient circular economy.

Reviewer 3 Report

The presented work is undoubtedly of scientific interest, and touches on an important topic - the development of composites based on biopolymers (PLA, PHA), reinforced with cellulose microfibers (MF). The authors show that the effectiveness of MF is not inferior to such a filler as cellulose nanocrystals (CNC).

I hope that the authors will take into account some of the comments and make appropriate corrections.

There are no references to figures and tables in the text of the article. Instead of links, there is the phrase “Error! Reference source not found "(for example, lines 115, 186, 222,254, 259 ... etc.). Most likely, this is a technical error; I recommend that authors check the manuscripts more carefully when uploading to the publisher's website.

Line 91 Delete “2. Results "

Lines 92-94 Delete paragraph

Lines 346-356, lines 372-373 You should format the text

Line 109 Could the authors describe in more detail the process of preparing composite films? (In my opinion, point 2.2 can be divided into two parts - preparation of samples with CNC-filled samples, and preparation of samples with MF, since the preparation methods have some differences). Is there a stage of removing the solvent from the films in this process?

Why did the authors choose exactly such amounts of MF - 10, 20 and 30 wt%? Are data available for 1, 2 and 3%? From what percentage of filler (in the case of MF) can the effect (increase in strength) be observed?

How do the authors compare the crystallinity of films with mechanical properties? (for example, in the case of PHA and PHA + 1% CNC crystallinity is 8.1 and 7.2, while Young's modulus is 0.54 and 0.72 GPa, respectively)

Do the authors have X-ray structural analysis data, do they correlate with DSC data?

Figure 5. No explanations for figures e-f

Figure 6 In figures “c” and “f” it is not possible to see the filler (20% MF), although it is clearly present in the optical photos (Fig. 5).

Author Response

There are no references to figures and tables in the text of the article. Instead of links, there is the phrase “Error! Reference source not found "(for example, lines 115, 186, 222,254, 259 ... etc.). Most likely, this is a technical error; I recommend that authors check the manuscripts more carefully when uploading to the publisher's website.

This error must be originated by the manipulation of the original document because of the incompatibility of the text processor between the authors and edition staff. All these notes are related to Microsoft Word Cross References and have been addressed.

Line 91 Delete “2. Results "

The text had been deleted. It had not been generated by the authors.

Lines 92-94 Delete paragraph

The text had been deleted. It had not been generated by the authors.

Lines 346-356, lines 372-373 You should format the text

This error must be originated by the manipulation of the original document because of the incompatibility of the text processor between the authors and edition staff. All these notes are related to Microsoft Word Cross References and have been addressed

Line 109 Could the authors describe in more detail the process of preparing composite films? (In my opinion, point 2.2 can be divided into two parts - preparation of samples with CNC-filled samples, and preparation of samples with MF, since the preparation methods have some differences). Is there a stage of removing the solvent from the films in this process?

The suggestion has been accepted and two subdivisions have been created to make a difference between the two types of samples. The removal of the solvent took place through evaporation at ventilated room conditions. This has been included in the manuscript.

Why did the authors choose exactly such amounts of MF - 10, 20 and 30 wt%? Are data available for 1, 2 and 3%? From what percentage of filler (in the case of MF) can the effect (increase in strength) be observed?

The mass fractions of MF (up to 30wt.%) were chosen since most of the authors traditionally work in this range of micro filler addition. Besides, higher MF mass fractions, given the physical properties of this type of filler, hindered the feeding, so the homogenization of the extruded material was not achieved. Mass fractions below 5wt.% were initially intended. However, the final mass fraction of the extruded material at these reduced filler content could not be precisely controlled, so insufficient accuracy was achieved for conclusive results. Moreover, this research was looking for high polymer replacement to reduce the cost of the biopolymer composites.

How do the authors compare the crystallinity of films with mechanical properties? (for example, in the case of PHA and PHA + 1% CNC crystallinity is 8.1 and 7.2, while Young's modulus is 0.54 and 0.72 GPa, respectively)

In this work, the authors only related the crystallinity of the different samples with the hygroscopic performance of the samples. It is conventionally accepted that the modification of the crystallinity influences the mechanical performance of the polymer. Nonetheless, the mechanical performance of a composite is also influenced by the mechanical properties of the filler, as it is noted in the exampled given by the reviewer. In this specific case, a sample with a reduced crystallinity exhibits a higher Young’s modulus. This is explained by the far superior Young’s modulus of CNC (>100GPa) that contributes to a higher Young’s modulus of the composite despite the reduced crystallinity induced by the addition of this filler.

Do the authors have X-ray structural analysis data, do they correlate with DSC data?

The authors did not consider the possibility of conducting X-ray tests (only X-ray diffractometer was available at our facilities) since this technique is not accurate enough for samples with great fractions of amorphous components. The background of the diffractograms of samples with low crystallinity may have a notorious hindrance in the analysis of the crystallinity due to the subjective criteria to be applied. Moreover, the lack of specific studies in this regard (explicit analysis of PHA crystallinity through XRD with clearly defined parameters) made that we rather focused on the exclusive analysis of DSC to determine the crystallinity of the polymers.

Figure 5. No explanations for figures e-f

After revising and addressing the problem detected by the corruption of Cross References, Figure 5e and Figure 5f appear commented in the text.

Figure 6 In figures “c” and “f” it is not possible to see the filler (20% MF), although it is clearly present in the optical photos (Fig. 5).

This is a remarkable observation. In Figure 6c, MF may not be visible since this surface is the one in contact with the atmosphere during solvent evaporation and that is why a porous surface is displayed. However, it is difficult to justify the reduced presence of MF in Figure 6f. In this fracture surface, only a few filaments are displayed. The only justification the authors may offer is the necking effect that reduces the cross-section under tensile loads. Thus, the polymer matrix is stretched and the filaments are pull-out from the matrix.

Round 2

Reviewer 3 Report

Lines 113, 388 Delete “Error! Reference source not found "

I would recommend the authors to modify Figure 7. The filler, the content of which is 20 wt.%, should be present in the photographs. Or the authors should add an explanation in the text of the article why they posted just such a photo.

The authors took into account most of the comments and made corrections. The data and results of the experiment in the article are presented appropriately. But in my opinion, this article is more appropriate for the journals "Materials" or "Processes", since the main attention is paid to the description of composite materials and their physical and mechanical characteristics. In any case, I leave the decision on publication to the editor-in-chief.

Author Response

Lines 113, 388 Delete “Error! Reference source not found "

Theses Error messages have been removed.

I would recommend the authors to modify Figure 7. The filler, the content of which is 20 wt.%, should be present in the photographs. Or the authors should add an explanation in the text of the article why they posted just such a photo.

The figure has been modified and the captions of the figure already include the information regarding the filler content